# A Retrospective, Single-Center Study Comparing Neoadjuvant ACTHP vs. DCbHP in HER2-Positive Early Breast Cancer Patients

**DOI:** 10.3390/cancers17020250

**Published:** 2025-01-14

**Authors:** Amit Itay, Opher Globus, Keren Levanon, Tal Sella, Rinat Bernstein-Molho, Tal Shapira, Cecilie Oedegaard, Dana Fourey, Einav Nili Gal Yam

**Affiliations:** 1Department of Oncology, Sheba Medical Center, Ramat Gan 52621, Israel; opher.globus@sheba.gov.il (O.G.); keren.levanon@sheba.gov.il (K.L.); tal.sella@sheba.gov.il (T.S.); rinat.bernstein@sheba.gov.il (R.B.-M.); tal.shapirarotenberg@sheba.gov.il (T.S.); cecilie.oedegaard@sheba.gov.il (C.O.); dana.fourey@sheba.gov.il (D.F.); 2Sackler Faculty of Medicine, Tel-Aviv University, Tel Aviv 69978, Israel

**Keywords:** Her-2 positive, breast cancer, neoadjuvant therapy, pathologic complete response, anthracyclines, congestive heart failure, early breast cancer, cardiotoxicity

## Abstract

For patients with stage II–III HER2-positive breast cancer, combination chemotherapy with dual anti-HER2 blockade in the neoadjuvant setting is a common standard. This study examined two drug combinations used for this purpose: one with anthracyclines and one without. Researchers compared their efficacy (pathologic complete response) and adverse effects, specifically cardiotoxicity, hospitalization and febrile neutropenia. Both treatments showed similar pathologic complete response rates; however, in the HER2 3+ subgroup, the anthracycline-containing regimen was statistically superior. The non-anthracycline regimen was less cardiotoxic.

## 1. Introduction

Locally advanced (T2–4 or node positive (N+)) human epidermal growth factor receptor 2 (HER2)-positive breast cancer is commonly treated with the neoadjuvant combination of cytotoxic chemotherapy plus dual antibody blockade—Trastuzumab/Pertuzumab (HP) therapy. Pathologic complete response (pCR) to systemic therapy in both tumor and lymph nodes correlates with improved survival and decreased risk of relapse [1,2]. A major concern related to anti-HER2 targeted therapy is congestive heart failure (CHF) [2,3,4,5]. This cardiotoxicity is often asymptomatic and, when symptomatic, mostly reversible [6,7]. Anthracyclines, on the other hand, may cause a dose and schedule-dependent irreversible CHF. The combination of anthracyclines and trastuzumab has been shown to cause long-lasting cardiac toxic effects in a subset of patients [7].

While various trials have examined the efficacy of anthracycline (A)-free regimens [8], only two prospective randomized trials have compared A-containing or A-free regimens combined with dual anti-HER2 blockade with HP.

The neoadjuvant, randomized phase 2 cardiac safety TRYPHAENA study compared three arms in which all patients received six cycles of neoadjuvant therapy every 3 weeks [9]. The first two A-containing arms included three cycles of 5-fluorouracil, epirubicin and cyclophosphamide (FEC) followed by three cycles of docetaxel (D), differing by the time of HP addition. The third A-free arm included six cycles of docetaxel and carboplatin (together with HP every 3 weeks (DCbHP). Adjuvant therapy was given to complete 1 year of trastuzumab. While not powered to formally compare pCR rates—the rates of pCR, defined as ypT0ypN0—were comparable, ranging between 45.3% and 51.9%. Left ventricular systolic dysfunction (LVSD), defined as a decline of more than 10% in left ventricular ejection fraction (LVEF) from baseline to <50%, was observed in 5.6%, 5.3% and 3.9% in the first, second and third arms, respectively (all grades). Only three patients (out of 24 with LVSD) were symptomatic. Long-term results from this study demonstrated similar three-year event-free survival (94% versus 93%) and overall survival (98%) for the A-free versus the A-containing regimens, respectively [10]. The TRAIN-2 trial [11] was an open-label, randomized, controlled, phase 3 trial comparing nine cycles of A-containing (3 FEC cycles followed by six cycles of Paclitaxel and Carboplatin) versus nine A-free cycles of Paclitaxel plus Carboplatin. HP was administered concurrently from the first cycle in both arms. Both arms showed similar pCR (ypT0/is ypN0) rates (67% vs. 68%, respectively, *p* = 0.95). Cardiac toxicity was low in both arms, and LVSD was observed in 5% of patients in the A-containing group vs. 3% in the A-free group (*p* = 0.32). Symptomatic LVSD was reported in two patients in the A-containing group. Febrile neutropenia (grade 3 or 4) was more prevalent in the A-containing arm, 10% vs. 1% in the A-free arm [9,11].

Notably, while both studies showed no pCR difference between the A-containing versus the A-free arms, pCR rates were numerically higher for A-containing regimens in both trials [9,11]. Importantly, both studies compared the less commonly used chemotherapy regimens (nine cycles of chemotherapy in the TRAIN-2 trial or only six cycles of the A-containing regimen in Tryphaena).

Nonetheless, based on these and previous adjuvant trials, clinical practice guidelines currently indicate A-free DCbHP as a preferred neoadjuvant regimen over the A-containing ACTHP [12].

As prospective and retrospective studies directly comparing the efficacy and safety of these two major regimens are lacking, we conducted this study comparing DCbHP to ACTHP in a single-center, real-world, retrospective cohort.

## 2. Materials and Methods

### 2.1. Study Population and Treatment

We performed a single-center, real-world, retrospective study including patients treated at the Sheba Medical Center between 9/2017 and 6/2022. The data were collected from the electronic medical records with the aid of the MDClone^TM^ platform to identify female patients with clinical stage II–III HER2-positive breast cancer who received neoadjuvant treatment. Subjects were considered to have HER2-positive cancer if the HER2/neu-immunohistochemistry (IHC) score was either +3 or +2 with FISH (fluorescence in situ hybridization) according to College of American Pathologists (CAP) guidelines. Estrogen receptor (ER) positive is defined as positive IHC staining in more than 1% of cells. Patients received either one of the following pre-operative regimens: (1) DCbHP—6 cycles of Docetaxel (75 mg/m^2^) plus Carboplatin AUC (Area under curve) 5 or 6 mg/mL*min every 3 weeks with prophylactic G-CSF (granulocyte-colony-stimulating factor) support concomitantly with Trastuzumab 8 mg/kg and Pertuzumab 840 mg loading dose in the first cycle followed by continuous dose of Trastuzumab 6 mg/kg and Pertuzumab 420 mg continuously every 3 weeks. (2) ACTHP—four cycles of dose-dense Doxorubicin (60 mg/m^2^) and Cyclophosphamide (600 mg/m^2^) (ddAC) every 2 weeks with G-CSF support followed by 12 weekly cycles of paclitaxel (80 mg/m^2^), plus HP every 3 weeks. After completion of neoadjuvant treatment, patients underwent definitive breast surgery with either sentinel lymph node biopsy (SLNB) or axillary lymph node dissection (ALND), as indicated.

Between 9/2017 and 6/2022, we identified 182 consecutive patients who initiated one of the two regimens. Of these, two patients refused surgeries following treatment, and one did not continue therapy in our hospital (lost to follow-up). For the 179 eligible patients, demographic, clinical, outcome, and toxicity variables were collected from the electronic medical records. Cardiac function data were abstracted from echocardiogram reports and oncologists’ and cardiologists’ notes. The clinical stage at diagnosis and pathological stage at the time of surgery was based on the 7th edition of the American Joint Committee on Cancer Staging Criteria (AJCC). CR was defined as the absence of residual invasive disease (with or without ductal carcinoma in situ) in the breast and sampled axillary nodes (ypT0/isN0). Based on clinical notes, treatment-related toxicities were retrospectively defined according to Common Toxicity Criteria for Adverse Events, version 4.03. LVSD was defined as an LVEF decline of more than 10%, compared to baseline, or a drop below 50%. New York Heart Association (NYHA) Functional Classification was determined by the cardiologist during the LVSD. Additional endpoints included dose reductions and missed doses, hospitalization during treatment, and febrile neutropenia events.

### 2.2. Statistical Analyses

Descriptive analyses of safety and efficacy were used to summarize the data. A multivariate analysis was performed. After frequency analysis and data cleanup, demographic and clinical variables were collected, and protocols were compared by crosstabulation with Chi-Square and Fisher’s exact test. The *p*-value for significance was *p*-value < 0.05. Odds ratios were calculated for all independent variables to establish differences between DCbHP and ACTHP, and stepwise logistic regressions were performed to elucidate treatment responses among patients treated with either regimen. Statistical analyses were performed with SPSS [IBM SPSS Ver.28].

The hospital’s Helsinki ethics board approved this trial.

## 3. Results

Of the 179 patients, 106 received neoadjuvant ACTHP, and 73 received neoadjuvant DCbHP (N = 73). Notably, 53.2% (39) of patients received DCbHP with an initial dose of carboplatin AUC = 5. The mean age at diagnosis of the overall study population was 48.4 years in both groups (26–75 in the ACTHP group and 20–74 in the DCbHP, *p* = 0.98). All patients were female; all had good performance status (PS 0–1) and a left ventricular ejection fraction (LVEF) of at least 50% before initiation of neoadjuvant therapy. Most tumors, 60.9% (109), were ER positive, 79.3% (142) were HER2 +3 and 71.0% (125) had nodal involvement. ER status, HER2 IHC (2+ FISH positive or 3+) and nodal status were balanced between the two regimens. Patient characteristics are described in Table 1.

### 3.1. Efficacy

pCR occurred in 63.1% of patients (113/179) in the overall cohort. pCR rates for subgroups are detailed in Table 2.

pCR was numerically higher with ACTHP, 67.0% (71), compared with DCbHP, 57.5% (42). However, this was not statistically significant (OR 1.497, 95% CI 0.809–2.772, *p* = 0.129). In the HER2 +3 subgroup, pCR was significantly higher with ACTHP compared with DCbHP (80% v. 62.9%, *p* = 0.036). pCR rates were substantially lower in the HER2 2+ subgroup, with no significant difference between the regimens (26.9% and 27.3%, respectively). Other subgroups showed numerical advantage for the ACTHP vs. the DCbHP group without statistical significance (see Table 3).

### 3.2. Safety

In the neoadjuvant phase, ACTHP was generally better tolerated than DCbHP. The percentage of patients who needed dose reductions was significantly higher for DCbHP (54.8%) compared with ACTHP (24.5%) (*p* <0.001). In the DCbHP group, 53.2% (39 patients) started with an initial dose of carboplatin AUC = 5, of which 20.5% (8) had a further dose reduction (AUC 4 or less). With ACTHP, febrile neutropenia, asthenia and neuropathy were the main causes of dose reductions, whereas with DCbHP, most were due to asthenia and diarrhea. In the DCbHP group, 5.5% (four patients) had missed chemotherapy cycles (1–2 cycles). In the ACTHP group, 1.9% (2) had missed doses in the AC part, while 13.2% (14) of patients missed 1–2 of the last weekly paclitaxel due to neuropathy (though they all received a total paclitaxel dose of at least 800 mg/m^2^). In addition, 16.9% of patients (18) in the ACTHP group switched from paclitaxel to docetaxel due to neuropathy but completed the equivalent taxane dosage.

Only 6.8% (five patients) in the DCbHP group received the complete chemotherapy plan without any dose reduction (received Carboplatin AUC6) or missed doses compared to 66% (70) of patients in the ACTHP groups who completed the chemotherapy plan.

Hospitalization rates during the neoadjuvant phase were similar with both regimens (ACTHP 14.2% (15 patients) vs. DCbHP 13.7% (10 patients) in the DCbHP regimen). Febrile neutropenia was numerically more common in the ACTHP vs. DCbHP regimen (6.6% vs. 1.4%, respectively), though not statistically significant (*p* = 0.09).

### 3.3. Cardiac Safety

LVEF dynamics measured cardiac safety throughout the neoadjuvant and adjuvant phases of treatment. Per standard of care, all patients had a baseline LVEF of ≥50% and thereafter had echocardiograms performed every 3 months (or sooner if clinically indicated) throughout the neo/adjuvant treatment period. LVSD was observed in 6.6% (seven patients) in the ACTHP group vs. 0% in the DCbHP group (*p* = 0.043). The mean age of these patients was 48.9 years (range: 26–63). Two patients experienced LVSD during the second part of the chemotherapy (taxanes), and five patients during the adjuvant HP phase. None experienced LVSD during the anthracycline part. Among those who received adjuvant T-DM1, none had LVSD while on this drug. Of the seven patients, one had asymptomatic LVEF decline (NYHA I), three had mild symptoms (NYHA II), and the other three patients had marked limitations of physical activity and were classified as NYHA III. None had overt heart failure, and none were hospitalized due to LVSD. For five of the patients, HP treatment was delayed by 2 weeks to 4 months. HP treatment was discontinued for the remaining two patients who experienced LVEF decline following the completion of their chemotherapy and surgery. All patients with LVEF decline had a cardiologist consultation and started cardio-protective treatment (beta-blocker and/or ACE inhibitors). In all patients, LVEF returned to normal (normal or >50%) within 3–6 months, and all continued to receive cardio-protective medications. In all patients, LVEF returned to normal (normal or >50%) within 3–6 months, and all continued to receive cardio-protective medications (see Table 4).

## 4. Discussion

In recent years, there has been a shift towards non-anthracycline-containing chemotherapy regimens in the treatment of HER2-positive BC to decrease long-term adverse outcomes, such as cardiotoxicity and leukemia [13].

This study aimed to compare pCR rates, tolerability and early cardiac safety of the two commonly used neoadjuvant regimens in locally advanced HER2-positive breast cancer in a real-life population: the anthracycline-containing ACTHP versus the anthracycline-free DCbHP regimen. To the best of our knowledge, no formal comparison between these two regimes has been performed in the neoadjuvant setting in a prospective or retrospective trial.

The overall pCR rate in our cohort was 63%, similar to previous studies [9,11]. While there was a numerical advantage in favor of ACTHP (67% vs. 57.5%), it was not statistically different. However, among subgroups, in HER2 +3 patients, the difference between the regimens was statistically significant in favor of ACTHP (80% pCR vs. 62.95% in DCbHP *p* = 0.036). In HER2 + 2 FISH-positive patients, pCR rates were comparatively low at around 27% (*p* = 1.0). These lower pCR rates in HER2 + 2 FISH-positive patients are in line with previous studies [14,15,16]. There was no statistical difference among patients in other subgroups based on tumor, nodal or ER status. Several explanations may explain the superiority of ACTHP in our study: the type of chemotherapy (A-containing vs. A-free), the number of treatment cycles (8 vs. 6) and a lower dose intensity due to more dose reductions in the DCbHP group.

Regarding types of chemotherapy and the number of cycles: In TRAIN-2 and TRYPHAENA, there was no advantage to the A-containing regimens. However, while caution should be exercised when making comparisons across trials, it is notable that absolute pCR rates were quite different between the two trials, possibly related to the number of treatment cycles that were given: In TRYPHAENA [9], chemotherapy was administered for six cycles, and pCR rates in both breast and nodes (ypT0N0) were in the range of 45–50% in all arms. In TRAIN-2 [11], nine cycles of chemotherapy were administered with pCR rates in both arms of 67–68%.

In addition to the TRAIN-2, TRYPHAENA and the current cohort, several additional neoadjuvant trials were performed that included A-containing or A-free chemotherapy combinations with pertuzumab and trastuzumab (see Table 5). Looking at pCR and survival/recurrence outcomes of these regimens, there seems to be a trend towards higher pCR rates when 8–9 chemotherapy cycles were administered (range 61–68%) compared with six cycles (range 45–58%), irrespective of the inclusion of anthracyclines (see Table 5). This may imply that the number of chemotherapy cycles is a major factor contributing to the pCR rate in these patients [17].

Another contributor to the lower PCR rates in the DCbHP group may be the lower tolerability of this regimen compared with ACTHP. We observed a significantly higher rate of dose reductions in the patients receiving DCbHP (54.8% of patients compared with 24.5% of patients receiving ACTHP [*p* < 0.001]). This is despite an initial carboplatin dose of AUC5 in more than half of patients.

In terms of cardiotoxicity, LVEF declines were significantly higher in the ACTHP group, observed among 6.6% of patients, compared with none of the patients treated with DCbHP (*p* = 0.043). Of these, 5.7% had a symptomatic decline of LVEF. This is in line with the 5.5% rate reported in TRYPHAENA [9]. In TRAIN2, LVEF declines were rare (1%) in the anthracycline-containing arm, though patients received only three cycles of epirubicin, which is associated with less cardiotoxicity compared with doxorubicin. There were no LVEF declines in the anthracycline-free arm.

Hospitalization rates were similarly low in both groups. Febrile neutropenia was more common with ACTHP (6.6%), though the difference was not statistically significant. The rate of febrile neutropenia in the A-containing arm of TRAIN2 was higher (10%), likely because only 30% of patients received secondary prophylactic G-CSF, whereas all our patients were prescribed primary prophylactic G-CSF.

The limitations of this study include its retrospective design, small cohort size and the need for further follow-up to detect recurrences and assess their correlation with pCR rates.

## 5. Conclusions

The dose-dense ACTHP regimen used in this real-world cohort was more effective regarding pCR than DCbHP in the HER + 3 group, with significantly higher cardiotoxicity.

While pCR correlates with long-term outcomes, it is unclear whether this pCR benefit will have an impact on survival outcomes in our cohort, which are yet immature as median follow-up of the DCbHP group is less than 3 years. According to current guidelines, patients who did not achieve pCR (57% in the ACTHP group and 77.4% in the DCbHP group) were treated with adjuvant T-DM1, which is expected to improve survival rates [28]. In addition, long-term toxicities of anthracyclines may affect overall survival outcomes, as recently described in an update of the “long-term follow-up of the role of Anthracyclines in Early Breast Cancer Trials” [29].

Longer follow-up of this and similar cohorts is needed to provide additional insights into the role of anthracyclines in HER2-positive breast cancer.

## Figures and Tables

**Table 1 cancers-17-00250-t001:** Patient characteristics by treatment regimen.

	ACTHP No. (%)	DCbHP No. (%)	*p*-Value
Total	106 (59.2)	73 (40.8)	
Mean age	48.4	48.4	0.98
T stage			0.43
T = 1/2	75 (70.8)	56 (76.7)	
T = 3/4	30 (28.6)	17 (23.3)	
HER2 IHC status			0.125
HER2 +3	80 (75.5)	62 (84.9)	
HER2 + 2 (FISH pos)	26 (24.5)	11 (15.1)	
HR status			0.65
ER neg	40 (37.7)	30 (41.1)	
ER pos	66 (62.3)	43 (58.9)	
Nodal status			0.846
N pos	74 (69.8)	51 (69.9)	
N neg	32 (30.2)	22 (30.1)	

T—tumor size, N—node, pos—positive, neg—negative, FISH—fluorescence in situ hybridization.

**Table 2 cancers-17-00250-t002:** Overall pCR rates in subgroups (N = 179).

	Total Number of Patients per Subgroup, No.	pCR Rates in Every Subgroup No. (%)
Total	179	113 (63.1)
HER-2 + 3	142	103 (72.5)
HER-2 + 2 FISH pos	37	10 (27)
ER neg	70	60 (85.7)
ER pos	109	53 (48.6)
N pos	125	72 (57.6)
N neg	51	38 (74.5)

pCR—pathologic complete response; ER—estrogen receptor, N—node, pos—positive, neg—negative, FISH—fluorescence in situ hybridization.

**Table 3 cancers-17-00250-t003:** pCR rates of ACTHP and DCbHP in clinically relevant subgroups (multivariate analysis).

	Therapy Protocol				*p*-Value [2-Sided]	OR [Odds Ratio]	95% CI Confidence Interval
	ACTHP		DCbHP				
	No.	No. with pCR (%)	No.	No. with pCR (%)			
Total	106	71 (67)	73	42 (57.5)	0.129	1.50	0.8–2.77
HER2 + 3	80	64 (80)	62	39 (62.9)	0.036	2.36	1.11–5.00
HER2 + 2 FISH pos	26	7 (26.9)	11	3 (27.3)	1.0	0.98	0.201–4.79
ER neg	40	37 (92.5)	30	23 (76.7)	0.09	3.75	0.88–15.99
ER pos	66	34 (51.5)	43	19 (44.2)	0.56	1.34	0.62–2.90
N pos	74	46 (62.2)	51	26 (50.9)	0.27	1.58	0.77–3.25
N neg	31	24 (77.4)	20	14 (70)	0.74	1.47	0.41–5.25
T1–2	75	50 (66.7)	56	29 (51.8)	0.09	1.86	0.92–3.79
T3–4	30	20 (66.7)	17	13 (76.5)	0.48	0.62	0.16–2.38

pCR—pathologic complete response; ER—estrogen receptor, N—node, T—tumor size, pos—positive, neg—negative, FISH—fluorescence in situ hybridization.

**Table 4 cancers-17-00250-t004:** Adverse events.

	ACTHP No (%)	DCbHP No (%)	*p*-Value	OR [95% CI]
LVEF reduction	7 (6.6)	0 (0)	0.043	11.08 * [0.6229–197.1]
Hospitalization	15 (14.2)	10 (13.7)	0.835	1.197 [0.522–2.74]
Febrile neutropenia	7 (6.6)	1 (1.4)	0.1441	5.091 [0.8532–58.07]

OR—odds ratio; LVEF—left ventricle ejection fraction. * Woolf logit correction.

**Table 5 cancers-17-00250-t005:** Trials in the neoadjuvant setting.

	Regimen	No of pts.	Total Treatment Cycles	pCR pT0/is ypN0 Unless Otherwise Stated	DFS/EFS (%)	OS (%) [95% CI]
Tryphaena [9,10]	FECHP → THP FEC → THP DCbHP	73 75 77	6 6 6	50.7% 45.3% 51.9% ypT0/ypN0	87 88 87 DFS (3 y)	94 94 93 (3 y)
Berenice [18,19]	AC → THP FEC → THP	199 201	8 8	62% 61%	EFS (5 y) 91 EFS (5 y) 89	96.1 (93.3–98.9) 93.8 (90.3–97.2) (5 y)
TRAIN-2 [11,20]	FEC → TCbHP TCbHP (X9)	211 206	9 9	67% 68%	EFS (3 y) 93 EFS (3 y) 94	98 (95.7–99.7) 98 (96.4–100) (3 y)
PHERgain [21,22]	TCbHP	71	6	58%	DFS (3 y) 98	N/A
Gepar-Septo [23,24] (For the 396 HER2-pos patients in the trial	TaxaneHP → ECHP	396	8	66%	N/A	N/A
Neopeaks [25,26]	DCbHP arm	51	6	57%	80.7 DFS (5 y)	97.5 (94.1–98.9) (5 y)
KRISTINE [27]	DCbPH	221	6	55.7%	iDFS (3 y) 94	N/A

H—trastuzumab; P—pertuzumab; T—paclitaxel; A—doxorubicin; C—cyclophosphamide;F—fluorouracil; E—epirubicin; D—docetaxel; Cb—carboplatin, pCR—pathologic complete response, DFS—disease-free recurrence, EFS—event-free recurrence, OS—overall survival, CI—confidence interval; N/A—Not applicable; y-year; ypT0/is ypN0—the absence of invasive disease in the breast and axilla—Chevallier’s criteria.

## Data Availability

Data available on request due to restrictions (ethical reasons).

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
