# Peer review of "A Retrospective, Single-Center Study Comparing Neoadjuvant ACTHP vs. DCbHP in HER2-Positive Early Breast Cancer Patients"

_cancers, 2025, doi:10.3390/cancers17020250_

Round 1

Reviewer 1 Report

Comments and Suggestions for Authors

Itay and colleagues performed a retrospective cohort study to evaluate the efficacy and safety of a combinatorial therapies with either DCbHP (Docetaxel, carboplatin, trastuzumab pertuzumab) or ACTHP (Doxorubicin & cyclophosphamide followed paclitaxel trastuzumab pertuzumab), in 106 and 73 stage II-III HER2-positive breast cancer patients, respectively. To examine the cardiac toxicity, they compared Left Ventricular Ejection Fraction (LVEF) decline of 120 more than 10%, compared to baseline, or a drop below 50% or cardiac death. Pathologic complete response (CR) was defined as the absence of residual invasive disease (with or without ductal carcinoma in situ) in the breast and sampled axillary nodes.  After completion of neoadjuvant treatment, when needed patients underwent definitive breast surgery with either sentinel lymph node biopsy (SLNB) or axillary lymph node dissection (ALND).

Many adjuvant chemotherapy combined with trastuzumab, (HER2 monoclonal antibody that inhibits HER2 dimerization), with pertuzumab (HERs monoclonal antibody that binds and inhibits HER2 heterodimerization with other HER family receptors), have markedly improved outcomes among patients with HER2-positive early breast cancer, reducing the risk of disease recurrence and death.  However, the patient's benefit must always be considered in relation to the magnitude of the side effects of the regimes, in particular, the anthracyclines-containing regimes. In this study, the investigators observed that 7 patients under ACTHP treatment had a substantial decrease in left ventricular ejection fraction (LVEF) reduction (p<0,043) with NYHA I, II or III. None (0%) was observed in patients receiving DCbHP (p<0.001). Other side effects, such as diarrhea, anemia, neutropenia, asthenia and neuropathy were better tolerated in patients under ACTHP treatment.  Although there was no statistically significant difference in pCR rates between all HER stage groups, the overall response to ACTHP (80%) was better in HER+3 subgroups concomitantly with higher cardiotoxicity. Overall, this study offers clinically important data to be confirmed in future investigations in a larger cohort of patients.

Reviewer 2 Report

Comments and Suggestions for Authors

In this study, the authors explored the efficacy and toxicity of two commonly used neoadjuvant regimens, ACTHP and DCbHP, in a real-life cohort. However, this study looks quite similar to many other recent studies that discuss the relationship between neoadjuvant regimens and breast cancer [1-3].

[1] Ferraro E, Singh J, Patil S, et al. Incidence of brain metastases in patients with early HER2-positive breast cancer receiving neoadjuvant chemotherapy with trastuzumab and pertuzumab. NPJ Breast Cancer, 2022, 8(1): 37.

[2] Lin B, Fan J, Liu F, et al. Efficacy and Safety of Dual Anti-HER2 Blockade and Docetaxel With or Without Carboplatin as Neoadjuvant Regimen for Treatment of HER2-Positive Breast Cancer. Technology in Cancer Research & Treatment, 2023, 22: 15330338231218152.

[3] Spring L, Niemierko A, Haddad S, et al. Effectiveness and tolerability of neoadjuvant pertuzumab-containing regimens for HER2-positive localized breast cancer. Breast cancer research and treatment, 2018, 172: 733-740.

Reviewer 3 Report

Comments and Suggestions for Authors

Comments on the Quality of English Language

Line 136, For the overall study population mean age at diagnosis was 48.4 years old, in both groups, (range: 26-75 in the ACTHP group and 20-74 in the DCbHP, P=0.98). This sentence needs to be rewrite. No subject. Please check the gramma for similar situation here across the whole manuscript.  

Round 2

Reviewer 3 Report

Comments and Suggestions for Authors

No further comments.